# Effectiveness of Oil-Based Denture Dentifrices-Organoleptic Characteristics, Physicochemical Properties and Antimicrobial Action

**DOI:** 10.3390/antibiotics10070813

**Published:** 2021-07-04

**Authors:** Andrezza C. M. dos Santos, Viviane C. Oliveira, Ana P. Macedo, Jairo K. Bastos, Mário S. Ogasawara, Evandro Watanabe, Isabela M. Chaguri, Cláudia H. Silva-Lovato, Helena F. O. Paranhos

**Affiliations:** 1Department of Dental Materials and Prosthodontics, School of Dentistry of Ribeirão Preto, University of São Paulo, Café Avenue S/N, Ribeirão Preto 14040-904, SP, Brazil; andrezzamoura@usp.br (A.C.M.d.S.); anapaula@forp.usp.br (A.P.M.); isabelamchaguri@gmail.com (I.M.C.); chl@forp.usp.br (C.H.S.-L.); helenpar@forp.usp.br (H.F.O.P.); 2Human Exposome and Infectious Diseases Network–HEID, School of Nursing of Ribeirão Preto, University of São Paulo, Bandeirantes Avenue 3900, Ribeirão Preto 14040-904, SP, Brazil; ewatanabe@forp.usp.br; 3Department of Pharmaceutical Sciences, School of Pharmaceutical Sciences of Ribeirão Preto, University of São Paulo, Ribeirão Preto 14040-904, SP, Brazil; jkbastos@fcfrp.usp.br (J.K.B.); ogasa@usp.br (M.S.O.); 4Department of Restorative Dentistry, School of Dentistry of Ribeirão Preto, University of São Paulo, Ribeirão Preto 14040-904, SP, Brazil

**Keywords:** complete denture, acrylic resin, biofilms, denture cleansers, dentifrices, oils, antimicrobial action, adverse effects

## Abstract

Denture dentifrices must be effective and not deleterious to prosthetic devices. This study formulated and evaluated dentifrices based on oils of *Copaifera officinalis*, *Eucalyptus citriodora*, *Melaleuca alternifolia*, *Pinus strobus,* and *Ricinus communis*. Organoleptic characteristics (appearance, color, odor, taste), physicochemical properties (pH, density, consistency, rheological, abrasiveness, weight loss, and surface roughness) and antimicrobial (Hole-Plate Diffusion–HPD)/anti-biofilm (Colony Forming Units–CFU) action against *Staphylococcus aureus*, *Streptococcus mutans,* and *Candida albicans* were evaluated. Formulations were compared with water (negative control) and a commercial dentifrice (positive control). The data were analyzed by Kruskal-Wallis and Dunn tests (α = 0.05). The organoleptic and physicochemical properties were adequate. All dentifrices promoted weight losses, with high values for *C. officinalis* and *R. communis*, and an increase in surface roughness, without differing from each other. For antimicrobial action, *C. officinalis* and *E. citriodora* dentifrices were similar to positive control showing effectiveness against *S. mutans* and *C. albicans* and no dentifrice was effective against *S. aureus;* regarding the anti-biofilm action, the dentifrices were not effective, showing higher CFU counts than positive control for all microorganisms. The dentifrices presented satisfactory properties; and, although they showed antimicrobial action when evaluated by HPD, they showed no effective anti-biofilm action on multispecies biofilm.

## 1. Introduction

Complete denture biofilm is composed of complex microbial communities arising from the association between oral and pathogenic microorganisms [1]. This biofilm must be removed daily, by proper cleaning since it can cause local and systemic diseases. Brushing is widely indicated for denture cleaning and has been considered a simple, inexpensive, and effective method [2,3,4,5,6,7]. However, the use of adequate dentifrices is essential to avoid adverse effects on prosthetic dental apparatus [8,9,10,11,12,13,14,15], and to ensure antimicrobial effectiveness, as it is one of the main factors that will promote biofilm removal [5,16,17,18,19,20,21]. Thus, suitable organoleptic and physicochemical properties, along with antimicrobial action, are prerequisites for dentifrices designed specifically for denture hygiene.

Dentifrices have a complex composition consisting of several agents, each of which have different functions. In addition to the abrasive agent, the disinfectant plays an important role in oral hygiene [17]. For this purpose, the literature has reported the possibility of using herbal products, since they have resulted in enhancing the antimicrobial, anti-biofilm, anti-inflammatory, and antioxidant action of these products [22,23,24,25,26,27]. Among the natural components used for dental application, antimicrobial activity has been demonstrated by the oils of *Bowdichia virgilioides Kunth, Copaifera officinalis*, *Curcuma longa*, *Eucalyptus citriodora*, *Melaleuca alternifolia*, *Pinus strobus,* and *Ricinus communis*, and these could be used as active principles in cleaning products [28,29,30,31,32,33,34]. Consequently, the incorporation of plant extracts into dentifrices has allowed the development of new formulations for oral hygiene, which are effective alternatives to the conventional types available. In vitro studies have evaluated dentifrices for natural teeth based on oils [22] and herbs [23] and have found antimicrobial efficacy against oral microorganisms associated with caries and periodontal disease. Clinical studies have reported that herbal dentifrices [26,27], as well as those containing melaleuca [27], and thymol, eugenol, and eucalyptus [35], oil-based dentifrices reduced the indexes of dental biofilm and gingivitis. As regards dentifrices for denture hygiene, Leite et al. evaluated the antimicrobial activity of *R. communis*-based experimental dentifrices and demonstrated that the formulation at 10% was efficient against bacteria and fungus, except for *S. aureus* [20]. Other studies concluded that the same formulation did not cause significant changes in surface roughness and color of artificial teeth [12], and in abrasiveness, hardness and color stability of resilient materials [14,15]. However, studies are still scarce.

Therefore, the aim of this study was to formulate and evaluate dentifrices for denture cleaning based on oils of *Copaifera officinalis*, *Eucalyptus citriodora*, *Melaleuca alternifolia, Ricinus communis,* and *Pinus strobus*. By means of an in vitro methodology, the following features of the above-mentioned dentifrices were evaluated: organoleptic characteristics (appearance, color, odor, and flavor), physicochemical properties (density, pH, consistency, rheological properties, and abrasiveness) and antimicrobial and anti-biofilm action against *Staphylococcus aureus*, *Streptococcus mutans,* and *Candida albicans*–microorganisms that are frequently present in denture biofilm and have demonstrated potential for pathogenicity [36,37,38,39,40,41]. The null hypotheses tested were that the experimental dentifrices would have adequate organoleptic and physicochemical properties, and antimicrobial action similar to that of commercial toothpaste, against the microorganisms tested.

## 2. Results

### 2.1. Organoleptic Characteristics and Physico-Chemical Properties

The organoleptic characteristics were classified as “normal” with no changes at the initial time (day 0) and after 15, 30, 60 and 90 days. The physicochemical properties are presented in Table 1 (density, pH, consistency, and rheological characteristics) and Table 2 (abrasiveness–weight loss and surface roughness).

There were significant differences between the groups for variation in mass and change in surface roughness (*p* < 0.001). When compared with the negative control, all dentifrices (positive control and experiments) promoted weight losses. The lowest mass loss was observed for positive control; and the highest mass losses occurred for the *C. officinalis* (*p* < 0.001) and *R. communis* (*p* < 0.001) dentifrices. All dentifrices were classified as having medium abrasiveness. All dentifrices promoted increase in surface roughness (*P. strobus p* < 0.001; *M. alternifólia p* = 0.001; *C. officinalis p* = 0.001; *E. citriodora p* = 0.002; *R. communis p* = 0.003; positive control *p* = 0.002), without differences between them.

### 2.2. Antimicrobial and Anti-Biofilm Activity

The results of antimicrobial action and anti-biofilm activity are presented in Table 3 and Table 4, respectively. Relative to antimicrobial activity, the dentifrices were not effective against *S. aureus*, but showed antimicrobial activity against *S. mutans* and *C. albicans* (Table 3). Significant differences were found between the groups (*p* < 0.001) for both microorganisms. The best results were observed for *C. officinalis* and *E. citriodora* which showed similar values to those of the positive control (Trihydral) for *S. mutans (p* = 0.988; *p* = 0.110) and *C. albicans* (*p* = 1.000; *p* = 1.000).

As regards anti-biofilm activity, significant differences were found between the groups (*p* < 0.001) for all microorganisms. For *S. aureus*, the positive control showed a higher value for reduction in the CFU counts than the experimental groups (*C. officinalis p* < 0.001; *P. strobus p* < 0.001; *M. alternifólia p* < 0.001; *E. citriodora p* < 0.001; *R. communis p* < 0.001), with no significant difference from the negative control (*p* = 0.268). For both *S. mutans* and *C, albicans*, no significant difference was found between experimental dentifrices and negative control. For these microorganisms only positive control promoted a reduction in biofilm rates. For *S. mutans*, positive control showed a higher value for reduction in CFU than the negative control (*p* < 0.001) and experimental dentifrices (*P. strobus p* < 0.001, *R. communis p* < 0.001, *M. alternifolia p* < 0.001, *C. officinalis p* = 0.001, *E. citriodora p* = 0.002). These results were also observed for *C. albicans* (negative control *p* = 0.001; *C.a officinalis p* < 0.001; *E.citriodora p* < 0.001; *R. communis p* = 0.002; *P. strobus p* = 0.027; *M. alternifólia p* = 0.028).

## 3. Discussion

In this study, dentifrices were formulated and evaluated with the aim of obtaining an effective and safe product. The microorganisms selected were those related to denture biofilm and played an important role in the development of oral pathologies [1]. Furthermore, Trihydral toothpaste (positive control) was used because it is indicated for natural teeth and prosthetic devices and has been shown to be effective against denture biofilm [17,18,20]. The null hypotheses were partially accepted, since all experimental dentifrices exhibited adequate organoleptic characteristics and satisfactory physico-chemical properties, but not all showed antimicrobial effectiveness against the microorganisms evaluated.

The results showed adequate organoleptic characteristics and no subsequent changes The physicochemical properties indicated that the dentifrices were suitable for use to clean dentures. The data found for density and consistency were within acceptable values for dentifrices [16,17]. The pH values (>7) showed the characteristic of neutral products and were within the range of 4.5 and 10.5, considered suitable for dentifrices. An acidic pH influences the viscosity and action of active principles [16,17]. Therefore, the results were favorable, because they allowed for a balanced oral pH, contributing to the maintenance of oral health, in addition to preventing damage to denture base acrylic resin. Relative to the rheological properties, the dentifrices showed low viscosity that could be considered suitable for denture dentifrices. The values obtained for the hysteresis area showed both moderate degree of thixotropy and rate of active principle release. This characteristic reflects the spreading power of the product and ensures good conditions of use and release of the active principle [17]. The yield value is defined by the shear stress, which is found in the rheogram graph, and allows the fluidity of the product to be defined. This datum is important relative to products recommended for denture cleaning, as it allows the product to be released from the storage tube with adequate viscosity and flow for use.

Highly abrasive dentifrices should not be used, as they can cause excessive wear and an increase in surface roughness of acrylic resin, making it difficult to remove biofilm [8,9,10,11,12,13,14,15,16]. All of the experimental dentifrices caused mass loss and were classified as having medium abrasiveness. The classification used is related to two years of simulated brushing with a brushing machine [10,11,13,42]. Therefore, the values obtained could be considered satisfactory, since a medium abrasiveness was obtained in a simulated brushing cycle of five years. Abrasive silica (Tixosil 73) has highly water-soluble particles and has low abrasive characteristics [17]. However, its association with the silica thickener (Tisoxil 43B) may lead to a small increase in abrasive action since there is interaction between the particles of both agents, with changes in consistency, and consequently, in the degree of abrasiveness [8,9]. The highest mass loss values observed for *C. officinalis* and *R. communis* dentifrices could be attributed to the oils since there were no other differences between the formulations. All dentifrices changed the surface roughness of the acrylic resin. These results agreed with the findings of previous studies that showed an increase in acrylic resin roughness with the use of brushing [8,9,10,11,12,13]. Therefore, brushing time may be the factor that influenced the magnitude of the values obtained.

Determination of the Minimal Inhibitory Concentration (MIC) by means of the broth microdilution method is frequently used for initial screening of antimicrobial activity. The results indicated that five oils had strong antimicrobial action against the microorganisms evaluated. Nonetheless, it is worth mentioning that this assay is not widely indicated for testing compounds of a lipophilic nature and other hydrophobic extracts because these compounds are incapable of completely diffusing in aqueous media [43]. Consequently, it has been suggested that the antimicrobial or anti-biofilm activity should be confirmed using additional assays. Therefore, in the present study, the capacity of the dentifrices for inhibiting microbial growth (Hole-Plate Diffusion) and removing mature biofilm from denture base acrylic resin specimens was assessed.

The denture biofilm is a complex community with an extremely variable composition [38,39]. The antimicrobial action of dentifrices was evaluated against microorganisms with potential for pathogenicity, commonly present in the oral microbiota and that has been isolated from the internal surface of complete dentures [19,36,37,39,40]. *C. albicans,* the most prevalent fungus in the human oral cavity, has been shown to be the primary cause of denture-related stomatitis [44,45,46,47,48,49,50]. *S. mutans* is responsible for consolidation and progression of dental biofilm, initial colonization of prosthetic surfaces, and its antigens have been related to denture-related stomatitis [51,52]. *S. aureus* is related to systemic infections, such as septicemia, endocarditis, pneumonia and abscesses [51,53,54], and to local diseases such as angular cheilitis, endodontic infections and mucositis [55,56].

With specific regard to *S. aureus*, taken as a whole, neither the antimicrobial nor the anti-biofilm evaluation indicated satisfactory results, a finding that was in disagreement with MIC results. The hole-plate diffusion assay has limitations due to the volatility, insolubility and irregular diffusion of oils [21]. This limitation could explain the variations in the antimicrobial results. Moreover, the literature has pointed that the susceptibility of microorganisms in states both associated and not associated with biofilm, is widely discrepant. The tolerance of biofilm to antimicrobial agents is about 100–1000 times greater when compared with that of the planktonic form [57]. This statement could explain the absence of anti-biofilm activity, therefore, formulations with high oil concentration might produce better results.

Studies have shown that *S. aureus* is resistant to the action of denture cleansers [54,58]. Whereas for yeasts, the expression of virulence factors and resistance genes may explain the complexity of controlling this microorganism [59]. The association of different active ingredients in peroxide solutions, such as potassium monopersulfate, sodium lauryl sulfate and titanium dioxide have been shown to enhance the antimicrobial action against this strain [60,61,62]. Likewise, the presence of sodium monofluorophosphate in a conventional dentifrice has been shown to act by inhibiting the enzymatic metabolism and adherence of this bacteria, and providing an antimicrobial effect [20,21]. According to the authors, the dentifrices may have interfered in the bacteria adhesion to the substrate and in organization of a polysaccharide matrix since the presence of surfactants and agents with antibiofilm activity can interfere in the surface tension of the substrate.

The antimicrobial findings regarding *S. mutans* and *C. albicans* indicated that the dentifrices showed eminent action, with better results for *C. officinalis* and *E. citriodora*. *C. officinalis* oil is considered an antibacterial agent against Gram-positive and Gram-negative pathogens present in the oral cavity [63,64], which makes it a promising substance with possibilities for the development of various oral care formulations [34,65]. This antimicrobial action may be attibutet to β-caryophyllene, the main bioactive constituent (58.7%) that alters membrane permeability and cell integrity, leading to membrane damage and intracellular content leakage [66]. *E. citriodora* oil has antimicrobial potential [67,68], and has been used in dental products [69]. Luqman et al. evaluated its action against several microorganisms and showed that this oil had more effective action against Gram-positive bacteria when compared with the Gram-negative types [31]. The eucalyptol, which accounted for approximately 79.5% of the *E. citriodora* oil, may be the responsible for this action, due to alteration on permeability and function of cell membrane leading to intracellular content leakage [70]. A low level of antimicrobial action of the *M. alternifolia* and *R. communis* dentifrices was also observed. The *M. alternifolia* oil has many applications in dentistry due its capacity for promoting alterations in the membrane permeability of yeast, consequently damaging the mitochondrial membrane, leading to cell death [24,30]. This mechanism of action is associated with the bioactive constituent terpinen-4-ol, which acts mainly on the cell membranes and organelles [71]. The antimicrobial action of *R. communis* oil is attributed to the toxicity of the protein ricin that inhibits the protein synthesis [72]. Solutions obtained from esterification of this oil have demonstrated moderate action against denture biofilm [40]; however, the full description of its mechanism of action has not yet been reported. The smallest halos were found for *P. strobus* dentifrice. Studies on *Strobus* species are scarce and its mechanism of action is not completely known. It has been suggested that its fungicidal action is related to high concentrations of hydrocarbon monoterpenes [29]. On this subject, α-pinene and β-pinene have been associated with important antimicrobial and antibiofilm actions [73]. Although promising antimicrobial action could be observed, the experimental dentifrices did not differ from the negative control and showed higher CFU counts than the positive control, without differing from each other. These results did not provide strong evidence that the bioactive constituents would improve the antibiofilm activity of oil-based dentifrices. Indeed, the variability in chemical composition of the oils is highly varied and the scientific literature is not conclusive about biological activities of all components. The effectiveness of Trihydral toothpaste is attributed to the presence of chloramine-T, an active ingredient capable of promoting oxidation and protein hydrolysis reactions [17,18,20]. In fact, the similarities between the values found for the negative control (brushing with water) and experimental groups could be explained by the force exerted by the mechanical brushing which even without dentifrice, acted by removing the biofilm.

Association of a toothbrush with a dentifrice is the most frequently used denture cleaning method [6,7]. However, further studies are still needed, since dentifrices for dentures have shown effective biofilm removal [2,3,4,5,16], but only moderate antimicrobial action [16,17,19,20]. Although natural products have been used in conventional dentifrices [22,23,24,26,27], their use in dentifrices for denture cleaning has not been widely reported.

Future studies should evaluate the effect of different associations between thickener, abrasive agents, and the proportion of humectant, in order to obtain formulations with a higher level of consistency and lower degree of abrasiveness. Furthermore, future analysis should consider the development of new formulations of dentifrices with increased oil concentration. This matter could clarify whether the absence of anti-biofilm activity could be associated with low oil availability. The present study was limited by the fact that the antimicrobial action was evaluated and observed against only one biofilm model composed of *S. aureus*, *S. mutans* and *C. albicans.* It is important to develop future studies with other microorganisms commonly found in the prosthetic biofilm. These formulations should also be tested in randomized clinical trials, in order to evaluate the factors related to the brushing process, and their efficacy against in vivo biofilm. These studies must include the use of a placebo dentifrice without an active ingredient, in order to allow evaluation of the action of the other constituents of the formulation, in an attempt to more clearly elucidate the antimicrobial action of the oils used.

## 4. Materials and Methods

### 4.1. Essential Oils and Fatty Acids

Oils were obtained from rhizomes, leaves, seeds or stems by steam distillation or cold pressing and their chemical compositions were determined by gas chromatography (Table 5).

### 4.2. Strains and Inoculum Preparation

Minus 80 °C glycerol stocks of *Streptococcus mutans* (ATCC 25175), *Staphylococcys aureus* (ATCC 25923) and *Candida albicans* (ATCC 10231) strains were thawed and streaked out on agar surface (*S. mutans*: Brain Heart Infusion–BHI (Kasvi, São José dos Pinhais, Paraná, Brazil); *S. aureus*: Tryptic Soy–TS (Kasvi); *C. albicans*: Sabouraud Dextrose–SD (Kasvi)). *S. aureus* and *C. albicans* plates were incubated at 37 °C for 24 h under aerobic conditions while *S. mutans* plates were incubated under microaerophilic conditions. Subsequently, one colony was transferred into 15 mL of broth medium and re-incubated at 37 °C for 19–24 h in order to obtain cells in an exponential growth phase. Then, the culture was centrifuged (Eppendorf, Hamburg, Germany) at 4200 *g* for 5 min and washed twice in phosphate buffered saline (PBS). The bacteria concentration per milliliter of PBS was determined by reading the optical density (OD) in a spectrophotometer (Thermo Scientific, Waltham, MA, USA), at the wavelength of 625 nm. Yeast cells were counted in a Neubauer chamber (HBG Company, Giessen, Germany).

### 4.3. Determination of Minimum Inhibitory Concentration (MIC)

For antimicrobial activity screening, the minimum inhibitory concentration (MIC) of the seven oils was determined by the broth microdilution method according to the Clinical and Laboratory Standards Institutes [74]. Five percent solutions were initially prepared in 5% (v/v) of dimethylsulfoxide (DMSO) (Sigma Aldrich, San Luis, MO, USA). Subsequently, two-fold serial dilutons of the oil-containing solutions were made in culture broths (BHI, TS and SD), in order to produce ten oil concentrations, ranging from 2.5 to 0.0048% (v/v). One hundred microliters of each concentration were placed into 96-well plates, in duplicate, (TPP, Trasadingen, Switzerland). Then, 10 μL of microorganism suspension (10^7^ CFU/mL for bacteria; 10^5^ CFU/mL for yeast) were added to each serial dilution and incubated at 37 °C for 24 h. Positive control and negative controls, respectively, were obtained by adding, or not adding the standardized microbial inoculum to culture medium without oil supplement. An additional control with two-fold serial dilutions of DMSO were prepared in order to verify the effect of the diluent on inhibiting the microorganism growth.

Microbial growth was evaluated by turbidity or absence of turbidity the culture medium. The MIC was determined as being the lowest concentration of the oil or DMSO to result in no microbial growth (Table 6).

To confirm growth inhibition, 20 μL of the suspension from each well were dropped onto a BHI, TS and SD agar surface. Microbial inhibition was categorized as strong (MIC < 0.5 mg/mL), moderate (0.5 ≤ MIC ≤ 1.5 mg/mL), and weak (MIC > 1.5 mg/mL) as previously reported [75].

The *C. officinalis*, *E. citriodora*, *R. communis*, *M. alternifólia* and *P. strobus* oils promoted strong growth inhibition of *S. aureus* and *S. mutans.* The *E. citriodora* and *M. alternifólia* essential oils promoted strong growth inhibition of *C. albicans*. Since these oils presented relevant antimicrobial activity, they were selected for the formulation of experimental dentifrices.

### 4.4. Dentifrice Formulations

Five formulations of dentifrices at final concentrations of 0.5% (v/v) of the oils were obtained. This concentration was used because it was considered feasible to obtain adequate formulations relative to the organoleptic characteristics, physicochemical properties and antimicrobial action. The dentifrices were prepared according to previously mentioned methodology [20]. Briefly, hydroxyethyl cellulose, glycerin, EDTA, saccharin sodium, and water were homogenized and kept at rest until gel formation. After this, the other components were added and mixed with the gel. After obtaining a homogeneous dentifrice, it was dispensed and stored in appropriated tubes. The dentifrice compositions are presented in Table 7.

### 4.5. Organoleptic Characteristics and Physico-Chemical Properties

After the dentifrices were obtained, the organoleptic characteristics were evaluated in time intervals of 15, 30, 60, and 90 days after their initial assessment [20]. The dentifrices were stored in white enameled aluminum tubes, protected in a humidity-free space and away from temperature variations. The appearance and color were observed by the visual method. Odor and taste were evaluated by olfactory and gustative methods. The odor and flavor analyses were performed considering a menthol-flavored product. Appearance was classified according to the criteria: I) Normal, II) Slightly separated, III) Separated. Color, odor and taste were classified according to the criteria: I) Normal, II) Slightly modified, III) Modified and IV) Intensely modified.

The physicochemical properties were evaluated in accordance with previous methodology [17]. The density was obtained by the equation d = m/v, where “d” is density, “m” is the measured mass and “v” the volume. Hydrogenionic potential was measured with a pHmeter (Digimed DM20; Digicrom Analytical Ltd.a., São Paulo, São Paulo, Brazil). Consistency was verified by the spreadability method, based on the sample flow under constant load in a pre-determined time interval. Rheological features were determined using a rheometer (Rheotest 2.0; VEB MLW Prufgerate–Werk Medingen, Sitz Freital, Germany). Abrasiveness was determined by evaluating the variation in mass, and surface roughness by the change in heat-polymerized acrylic resin specimens (90 mm × 30 mm × 3 mm; Clássico Produtos Odontológicos Ltd.a., São Paulo, SP, Brazil; *n* = 84) after simulated artificial brushing performed by a machine (Mavtec, Ribeirão Preto, SP, Brazil), in accordance with ISO/DTS 145691 specifications for wear testing [42]. The specimens were distributed (*n* = 12) into groups: brushing without dentifrice (only distilled water–23 ± 3 °C) (Negative Control) and six dentifrices–five experimental types (oils) and one commercial (Positive Control) (Trihydral-Perland Pharmacos, Cornelio Procópio, PR, Brazil). The machine worked at a rate of 356 rpm, under a 200 g load and a linear cleaning movement length of 3.8 cm. Each specimen received 10 mL of the suspension (distilled water or dentifrice diluted in distilled water at ratio of 1:1) and was brushed with soft toothbrushes (Tek; Johnson & Johnson, São José dos Campos, São Paulo, Brazil). The brushing time was 250 min (89,000 cycles), corresponding to five years of simulated exposure to brushing [10]. Suspensions and brushes were replaced at each time interval of 50 and 100 min, respectively. Before and after the test, the specimens were weighed and the variation in mass (mg) was obtained, and classified as: low (values up to 24 mg); medium (from 25 to 45 mg) and high (values over 46 mg) [10]. The surface roughness measurements were also obtained (µm; three readings of 4.0 mm long, 0.8 mm cut-off and at 0.5 mm/s) by means of a roughness tester (Surftest SJ-201P, Mitutoyo Corporation, Kawasaki, Japan). The arithmetic average of the three measurements was calculated.

### 4.6. Antimicrobial and Anti-Biofilm Activity

The antimicrobial activity was estimated by measuring zones of inhibition by the hole-plate diffusion method (HPD) [22] against *S. mutans*, *S. aureus,* and *C. albicans*. In parallel, the anti-biofilm activity was assessed against a model of multispecies biofilms, composed of the same microorganisms [62,76] and in accordance with Paranhos et al. [61]. The assays were conducted in three different time intervals.

For the hole-plate diffusion method, agar culture media (BHI, TS an SD) were prepared, sterilized and dispensed into 90 mm^2^ sterile Petri dishes to provide a base layer of 8 mL. Microbial inoculums with 10^6^ CFU/mL of each microorganism were added to aliquots of 12 mL culture media (45 °C). The suspensions obtained were deposited onto a base layer. After gelation, plastic straws were used to make three holes measuring 5.0 mm in diameter in each Petri dish. These holes received ~20 μL of each of the 06 dentifrices. The Petri dishes were pre-incubated at ambient temperature (25 °C) for 2 h, to allow the product to diffuse into the culture medium. After the pre-incubation period had elapsed, the Petri dishes were incubated in a microbiological oven at 37 °C for 24 h. The microbial growth inhibition halos that formed around the dentifrices were measured with the aid of a millimeter ruler, at three different points. in order to obtain a mean value. The measurements (mm) were made at the longest distance between two points, (from the outer limit of the inhibition zone to the hole). As this test was performed in triplicate, at three different time intervals, nine halos were obtained, and consequently, 9 measurements for each microorganism.

For anti-biofilm activity, 126 specimens (15 × 3 mm) of heat-polymerized acrylic resin (Clássico, Artigos Odontológicos Ltd.a., São Paulo, São Paulo, Brazil) were sterilized by means of microwave irradiation (650 W, 6 min) [60,61,62] and randomly distributed in 24-well tissue culture plates (TPP). Each well received 2 mL of BHI Broth (BHIB) inoculated with 10^6^ CFU/mL of *C. albicans* and 10^7^ CFU/mL of *S. aureus* and *S. mutans*. The plates were incubated at 37 °C for 90 min, at 75 rpm, under microaerophilic conditions (adhesion period). After this, each specimen was washed twice with PBS to remove non-adherent microorganisms. To promote biofilm growth and maturation, 2 mL of fresh BHIB was added to each well. Then, the plates were incubated at 37 °C for 48 h, at 75 rpm, under microaerophilic conditions. With the aim of confirming asepsis of the procedures, two specimens were not inoculated, and received only sterile culture media. Afterwards, specimens were distributed (*n* = 18) into the seven groups previously described and submitted to mechanical brushing. The specimens were fixed in sterile polymethyl methacrylate plates (Plexiglass; Day Brazil, Ribeirão Preto, São Paulo, Brazil), and brushed with the dentifrices in an artificial brushing machine, as previously described, for 3 min. (16.2 cycles), corresponding to three daily brushing events of one minute. After this, each specimen was aseptically removed, washed three times with PBS and transferred to tubes with 10 mL of Letheen broth (BD Difco, Sparks, MN, USA). After sonication (200 W; 40 KHz) (Altsonic, Ribeirão Preto, São Paulo, Brazil) for 20 min, serial dilutions (10^−1^ to 10^−4^) of the resultant suspension were seeded onto Petri dishes containing selective culture medium [*S. mutans*: Mitis Salivarius Agar (HiMedia, Mumbai, India) supplemented with 100 U/mL of Nystatin (Sigma Aldrich, Saint Louis, MO, USA), 0.2 U/mL of Bacitracin (Sigma Aldrich) and 20% (w/v) of sucrose (Dinâmica, Indaiatuba, SP, Brazil); *S. aureus*: Mannitol Salt Agar (Kasvi) supplemented with 100 U/mL of Nystatin; *C. albicans*: Sabouraud Dextrose Agar (Kasvi) supplemented with 0.05 g/L cloranfenicol (Sigma)]. Then, the Plates were incubated in a microbiological incubator at 37 °C for 48 h. *S. mutans* plates were incubated under microaerophilic conditions. After the incubation period, the number of colonies was counted, the value of colony forming units (CFUs) was obtained and transformed to log_10_(CFU + 1).

### 4.7. Statistical Analysis

The organoleptic characteristics were presented by descriptive analysis. Values relative to density, pH, consistency, viscosity and hysteresis area are presented in Table. After assumption of non-normal distribution (Shapiro–Wilk test) and non-homogeneous variances (Levene test), the data involving abrasiveness and antimicrobial action were submitted to the Kruskal-Wallis followed by the Dunn post-hoc test with Bonferroni adjustment (α = 0.05). All statistical tests were performed by a blinded researcher using the IBM SPSS Statistics for Windows 21.0 software (IBM Corp.).

## 5. Conclusions

Based on the methodology used and the results obtained, within the limitations of this study, it was concluded that the experimental oil-based denture dentifrices presented satisfactory organoleptic and physicochemical properties. Nonetheless, the results clearly illustrated that none of the dentifrices evaluated was capable of significantly reducing the multispecies biofilm viability.

## Figures and Tables

**Table 1 antibiotics-10-00813-t001:** Physical-chemical properties of experimental dentifrices.

Dentifrice	Density (g/mL)	pH	Consistency (mm)	Viscosity		Hysteresis Area
				Curve Ascending	Curve Downward	
*C. officinalis*	1.067	7.32	89.6	3692.58	124,222.22	0.66
*E. citriodora*	1.111	7.36	94.0	3976.63	155,277.78	1.21
*R. Communis*	1.116	7.36	93.4	3408.54	155,277.78	0.15
*M. alternifolia*	1.106	7.35	89.6	3195.50	155,277.78	0.97
*P. strobus*	1.075	7.37	81.0	3124.49	124,222.22	1.74

**Table 2 antibiotics-10-00813-t002:** Descriptive statistics of mass loss (mg), variation in roughness (ΔRa–μm) and statistical comparisons.

Properties	Group	Mean ± SD (Median)	95% CI (Range)	*p* Value
Mass Loss (mg)	*C.a officinalis*	−41.0 ± 4.1 (−41.7) ^d^	−43.6; −38.3 (−46.7; −34. 2)	<0.001 *
	*E. citriodora*	−35.6 ± 5.2 (−36.2) ^cd^	−38.9; −32.3 (−42.7; −23.6)	
	*R. communis*	−40.3 ± 5.6 (−40.2) ^d^	−43.9; −36.8 (−48.1; −28.0)	
	*M. alternifólia*	−37.2 ± 3.6 (−37.8) ^cd^	−39.4; −34.9 (−42.1; −29.8)	
	*P. strobus*	−33.3 ± 4.9 (−32.2) ^c^	−36.4; −30.2 (−42.2; −26.2)	
	Negative Control	2.6 ± 1.1 (2.7) ^a^	1.9; 3.3 (1.2; 4.2)	
	Positive Control	−24.3 ± 5.4 (−24.1) ^b^	−27.7; −20.8 (−30.7; −14.6)	
ΔRa (μm)	*C. officinalis*	3.86 ± 3.98 (1.35) ^b^	1.33; 6.39 (0.28; 11.37)	<0.001 **
	*E. citriodora*	2.97 ± 2.75 (1.93) ^b^	1.22; 4.72 (0.55; 9.51)	
	*R. communis*	2.88 ± 2.03 (3.05) ^b^	1.58; 4.17 (0.40; 5.77)	
	*M. alternifólia*	3.50 ± 2.99 (2.50) ^b^	1.59; 5.40 (0.41; 8.35)	
	*P. strobus*	5.40 ± 3.50 (5.56) ^b^	3.18; 7.62 (1.27; 12.41)	
	Negative Control	0.01 ± 0.02 (0.01) ^a^	−0.01; 0.02 (−0.04; 0.04)	
	Positive Control	3.17 ± 2.66 (2.69) ^b^	1.48; 4.86 (0.52; 9.25)	

SD: Standard deviation; CI–Confidence Interval for Mean; Range (minimum; maximum); Negative Control: brushing without dentifrice (water); Positive Control: Trihydral Commercial toothpaste; * ANOVA; ** Kruskal-Wallis test; ^abcd^ equal letters indicate statistical similarity (*p* > 0.05).

**Table 3 antibiotics-10-00813-t003:** Descriptive statistics of inhibition halo (mm) for *S. mutans* and *C. albicans* and statistical comparisons.

Microorganisms	Group	Mean ± SD (Median)	95% CI (Range)	*p* Value *
*S. mutans*	*C. officinalis*	1.3 ± 0.8 (1.8) ^abc^	0.7; 1.9 (0.0; 1.8)	<0.001
	*E. citriodora*	1.5 ± 0.7 (1.9) ^bc^	0.9; 2.0 (0.0; 1.9)	
	*R. communis*	0.8 ± 0.4 (0.9) ^ab^	0.4; 1.1 (0.0; 1.2)	
	*M. alternifólia*	0.3 ± 0.4 (0.0) ^ab^	0.0; 0.6 (0.0; 0.9)	
	*P. strobus*	0.2 ± 0.3 (0.0) ^a^	0.0; 0.4 (0.0; 0.8)	
	Positive Control	2.7 ± 0.3 (2.6) ^c^	2.5; 2.9 (2.4; 3.2)	
*C. albicans*	*C. officinalis*	1.4 ± 0.2 (1.5) ^bc^	1.2; 1.5 (1.0; 1.5)	<0.001
	*E. citriodora*	1.4 ± 0.3 (1.6) ^c^	1.2; 1.6 (0.9; 1.6)	
	*R. communis*	1.1 ± 0.1 (1.1) ^ab^	1.0; 1.2 (0.9; 1.3)	
	*M. alternifólia*	0.9 ± 0.2 (0.9) ^ab^	0.7; 1.0 (0.3; 1.1)	
	*P. strobus*	1.0 ± 0.1 (1.0) ^a^	1.0; 1.1 (0.9; 1.1)	
	Positive Control	1.3 ± 0.1 (1.3) ^c^	1.2; 1.4 (1.2; 1.5)	

SD: Standard deviation; CI–Confidence Interval for Mean; Range (minimum; maximum); Positive Control: Trihydral Commercial toothpaste; * Kruskal-Wallis test; ^abc^ equal letters indicate statistical similarity (*p* > 0.05).

**Table 4 antibiotics-10-00813-t004:** Descriptive statistics of log_10_ (CFU+1) for *S. mutans*, *S. aureus* and *C. albicans* and statistical comparisons.

Microorganisms	Group	Mean ± SD (Median)	95% CI (Range)	*p* Value *
*S. mutans*	*C. officinalis*	4.56 ± 0.82 (4.69) ^b^	4.15; 4.96 (2.60; 5.72)	<0.001
	*E. citriodora*	4.46 ± 0.99 (4.53) ^b^	3.97; 4.95 (1.61; 6.06)	
	*R. communis*	5.12 ± 0.77 (5.27) ^b^	4.74; 5.50 (3.60; 6.56)	
	*M. alternifólia*	4.88 ± 0.62 (4.92) ^b^	4.57; 5.19 (3.90; 5.91)	
	*P. strobus*	5.24 ± 0.74 (5.48) ^b^	4.88; 5.61 (3.60; 6.48)	
	Negative Control	5.06 ± 0.52 (5.06) ^b^	4.81; 5.31 (4.01; 5.97)	
	Positive Control	0.60 ± 1.20 (0.00) ^a^	0.00; 1.20 (0.00; 3.64)	
*S. aureus*	*C. officinalis*	6.20 ± 0.74 (6.38) ^c^	5.83; 6.56 (4.45; 7.08)	<0.001
	*E. citriodora*	6.09 ± 0.88 (6.14) ^bc^	5.65; 6.53 (3.66; 7.93)	
	*R. communis*	5.92 ± 0.56 (5.96) ^bc^	5.64; 6.19 (4.60; 6.63)	
	*M. alternifólia*	6.20 ± 0.54 (6.08) ^c^	5.93; 6.46 (5.30; 7.08)	
	*P. strobus*	6.20 ± 0.53 (6.34) ^c^	5.94; 6.46 (4.71; 6.91)	
	Negative Control	5.48 ± 0.47 (5.49) ^ab^	5.26; 5.71 (4.72; 6.46)	
	Positive Control	3.50 ± 1.21 (3.39) ^a^	2.90; 4.10 (1.61; 6.03)	
*C. albicans*	*C. officinalis*	3.29 ± 0.48 (3.34) ^b^	3.05; 3.53 (2.08; 4.17)	<0.001
	*E. citriodora*	3.00 ± 0.43 (2.89) ^b^	2.79; 3.21 (2.30; 3.77)	
	*R. communis*	2.90 ± 0.54 (2.92) ^b^	2.63; 3.17 (2.08; 4.09)	
	*M. alternifólia*	2.73 ± 0.41 (2.77) ^b^	2.53; 2.93 (1.91; 3.41)	
	*P. strobus*	2.58 ± 0.87 (2.76) ^b^	2.15; 3.01 (0.00; 3.73)	
	Negative Control	2.83 ± 0.68 (2.90) ^b^	2.50; 3.16 (1.61; 3.60)	
	Positive Control	1.47 ± 1.03 (1.61) ^a^	0.96; 1.98 (0.00; 2.95)	

SD: Standard deviation; CI–Confidence Interval for Mean; Range (minimum; maximum); Negative Control: brushing without dentifrice (water); Positive Control: Trihydral Commercial toothpaste; * Kruskal-Wallis test; ^abc^ equal letters indicate statistical similarity (*p* > 0.05).

**Table 5 antibiotics-10-00813-t005:** Chemical characterization of the oils.

Oils	Source	Manufacturer	Chemical Constituents *
*C. officinalis*	Stems	Oshadhi Brazil	Essential oil: β-caryophyllene (58.73%); α-humulene (7.81%); α-bergamothene (4.96%); α-copaene (4.66%); Germacrene (4.30%); ∆-cadinene (2.19%); β-selinene (1.73%); β-elemene (1.56%); α-cubebene (0.56%).
*E. citriodora*	Leaves	Sítio das Melaleucas, Ibiuna, SP, Brazil	Essential oil: Eucalyptol (79.53%); trans-β-ocimene (14.86%); o-cymene (1.57%); 6,6-dimethyl-2-methylene (1.06%); α-terpineol (0.66%); 4-methyl-1-(1-methylethyl) (0.57%); α-pinene oxyde (0.49); 6-octenal (0.47%); (R)-α-terpinyl acetate (0.42); β-myrcene (0.36%).
*M. alternifolia*	Leaves	Sítio das Melaleucas	Essential oil: Terpinen-4-ol (32.1%); y-terpinene (22.6%); α-terpinene (11.00%); terpinolene (4.00%); α-pinene (2.80%); viridiflorol (2.80%); α-terpineol (2.50%); 1,8-cineole (2.4%); β-gurjunene (2.10%); limonene (1.80%); p-cymene (2.20%); myrcene (0.9%); α-thujene (1.10%); β-pinene (0.90%); sabinene (0.90%).
*P. strobus*	Leaves	Oshadhi Brazil	Essential oil: α-pinene (33.02%); β-pinene (30.41%); myrcene (9.19%); limonene (9.16%); ∆3-carene (6.39%); caryophyllene (4.52%); terpinolene (1.24%); bornyl acetate (1.02%); β-caryophyllene (0.62%); α-terpineol (0.57%); α-Humulene (0.27%); bornyl (0.25%); δ-cadinene (0.25); Terpinen-4-ol (0.20).
*R. communis*	Seeds	Laszlo Group	Fatty acids: C18:1OH-ricinoleic (84.10%); C18:2-linoleic (4.60%); C18:1-oleic (3.60%); C16:0-palmitic (1.30%); C18:0-stearic (1.10%); C18:3-linolenic (0.60%); C22:0-behenic (0.60%); C20:0-arachidic (0.50%).

* According to manufacturer’s information.

**Table 6 antibiotics-10-00813-t006:** Minimum Inhibitory Concentration (MIC) of oils against *S. aureus*, *S. mutans* and *C. albicans*.

	Minimum Inhibitory Concentration (%)						
**Microorganisms**	*B. virgilioides Kunth*	*C. officinalis*	*C. Longa*	*E. citriodora*	*R. Communis*	*M. alternifolia*	*P. strobus*
*S. aureus*	>2.5	<0.009	2.5	0.62	<0.009	<0.009	<0.009
*S. mutans*	>2.5	0.0048	1.25	0.62	0.078	0.078	0.009
*C. albicans*	>2.5	>2.5	>2.5	0.62	>2.5	0.62	>2.5

**Table 7 antibiotics-10-00813-t007:** Basic composition of experimental dentifrices.

Components	Manufacturer	Function
Hydroxyethylcellulose	Fagron Rubber Industry Products Ltd.a, Guarulhos, SP, Brazil	Thickener
Glycerin	Ely Martins, Ribeirão Preto, SP, Brazil	Humectant
EDTA	Fagron P Rubber Industry Products Ltd.a, Guarulhos, SP, Brazil	Chelating Agent
Sodium benzoate	Labsynth Ltd.a, Diadema, São Paulo, SP, Brazil	Preservative
Cocamidopropyl betaine	Fagron Rubber Industry Products Ltd.a, Guarulhos, SP, Brazil	Surfactant
Oils	Laszo Group Oshadhi Brazil Sítio das Melaleucas	Antimicrobial active
Silica (Tisoxil 73)	Rhodia Solvay Group, São Paulo, SP, Brazil	Abrasive
Silica (Tisoxil 43 B)	Rhodia Solvay Group, São Paulo, SP, Brazil	Thickener
Titanium dioxide	Fagron Rubber Industry Products Ltd.a, Guarulhos, SP, Brazil	Pigment (white)
Menthol aroma	Givaudan of Brazil Ltd.a, São Paulo, SP, Brazil	Flavoring
Distilled water	-	Vehicle

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
