# Peer review of "Effectiveness of Oil-Based Denture Dentifrices-Organoleptic Characteristics, Physicochemical Properties and Antimicrobial Action"

_antibiotics, 2021, doi:10.3390/antibiotics10070813_

Round 1

Reviewer 1 Report

Dear Authors

This work is very fruitful for the future of Oil-based Denture Dentifrices. Denture stomatitis is the most common problem with denture wearers. Your work is good for future directions. 

I am suggesting a few critiques for the improvement of your writing and develop an interest in the readers. 

a) Abstract and keyword can be improved by authors. 

b) Line 46-50: I couldn't find something about the effectiveness of Oil-based denture denitrifies. Add some past published information. 

c) Why authors used the S.Aureaus strain for antimicrobial testing? Is any justification in the manuscript. 

d) Heading 4.6. Not well-presented authors can improve it and try to add recent publication from the specialized dental journal. In the past 5 years, a lot of studies has been presented on this topic. 

e) Authors have done a good presentation of the result part but scientific coherence has lacked. I would recommend taking expertise from a native English speaker or a additional suggestion for that heading. 

f) Add the limitation of this work faced by authors. 

Overall I would like to suggest improve this manuscript. 

Author Response

a) Abstract and keyword can be improved by authors. 

The abstract was reviewed and rewritten. The keywords were reviewed. Three keywords were added.

b) Line 46-50: I couldn't find something about the effectiveness of Oil-based denture denitrifies. Add some past published information. 

This information was added (Introduction - lines 60-71).

New references have been added to the reference list (highlighted in red).

c) Why authors used the aureus strain for antimicrobial testing? Is any justification in the manuscript. 

The justification was added (lines 78-79 – Introduction)

Two paragraphs were added on “Discussion” session (lines 199-208; and lines 219 - 229).

New references have been added to the reference list (highlighted in red).

d) Heading 4.6. Not well-presented authors can improve it and try to add recent publication from the specialized dental journal. In the past 5 years, a lot of studies has been presented on this topic.

References were added (22, 60-62 ) 

e) Authors have done a good presentation of the result part but scientific coherence has lacked. I would recommend taking expertise from a native English speaker or a additional suggestion for that heading.

The “Results” section was reviewed.

The manuscript was revised by an English language expert.

f) Add the limitation of this work faced by authors.

The limitations were described at lines 274 - 282. 

Overall I would like to suggest improve this manuscript. 

Reviewer 2 Report

The manuscript antibiotics-1268904 attempts to evaluate the effectiveness of dentifrices formulations based on essential oils, in terms of organoleptic characteristics, physicochemical properties and antimicrobial activity.

I have several comments on the manuscript, as follows:

My main concern is related to the essential oils used in the study.

For each species investigated, please provide information related to plant material that was used to isolate the essential oils (leaves, aerial part etc.).

Related to the isolation of essential oils, please describe the methodology used (type of apparatus, hydrodistillation time, essential oil yield).

The essential oils need to be characterized in terms of constituents (qualitative and quantitative analysis), therefore, please provide such information (gas chromatography analysis of essential oils needs to be undertaken).

More, correlation between the composition of essential oils and their antimicrobial activity should be undertaken. More, as the essential oils are the bioactive constituents of the dentifrices formulations, correlations between main constituents of essential oils and the observed antimicrobial activity of the formulations are highly demanded. One can not refer to literature data in general on the antimicrobial activity of the essential oil derived from a plant species; the variability in chemical composition is high, the constituents vary related to part of the plant being used, and more, to pedo-climatic conditions, storage, method of isolation of volatile compounds.

Regarding the MIC determination of the essential oils: as known, essential oils contain lipophilic compounds, please clarify how the serial dilutions were obtained and whether they were homogenous, and also if you used emulsifiers, please specify which and also state their concentration in the serial dilutions.

Overall, the manuscript should be accepted after Major revisions, attending the above-mentioned comments.

Author Response

For each species investigated, please provide information related to plant material that was used to isolate the essential oils (leaves, aerial part etc.). Related to the isolation of essential oils, please describe the methodology used (type of apparatus, hydrodistillation time, essential oil yield). The essential oils need to be characterized in terms of constituents (qualitative and quantitative analysis), therefore, please provide such information (gas chromatography analysis of essential oils needs to be undertaken).

The chemical characterization of the essential oils, as well as the information related to plant material was described on "Method" section (table 5; Lines 286 - 288).

In addition, the chemical compositions of the oils, determined by gas chromatography, are shown in table 5.

More, correlation between the composition of essential oils and their antimicrobial activity should be undertaken. More, as the essential oils are the bioactive constituents of the dentifrices formulations, correlations between main constituents of essential oils and the observed antimicrobial activity of the formulations are highly demanded. One can not refer to literature data in general on the antimicrobial activity of the essential oil derived from a plant species; the variability in chemical composition is high, the constituents vary related to part of the plant being used, and more, to pedo-climatic conditions, storage, method of isolation of volatile compounds.

This information was added on "discussion" section (Lines 253 – 260).

Regarding the MIC determination of the essential oils: as known, essential oils contain lipophilic compounds, please clarify how the serial dilutions were obtained and whether they were homogenous, and also if you used emulsifiers, please specify which and also state their concentration in the serial dilutions.

Dimethylsulfoxide (DMSO) at concentration of 5% was used to emulsifier the essential oils. This  information is stated at lines 310 – 311.

Reviewer 3 Report

The manuscript deals with an interesting topic that could lead to important industrial applications. However, the methodology used is not adequate for a study of the activity of essential oils.

The abstract should have the following order: background-introduction, objectives, methodology, results, discussion and conclusions. In the current form it is difficult to understand.

The introduction lacks information regarding the chemical composition and biological activity of the essential oils studied.

The methodology has a serious error that is difficult to solve. The essential oils used are not chemically characterized. This phase is essential for the study of biological activities. Plant essential oils depend on many factors:
In the first place would be the botanical origin, since each species has a chemical composition and even within the same species we can find several chemical races (chemotypes). The quantitative and qualitative characteristics of a species vary according to the phase of the vegetative cycle in which it is found. It also influences the qualities of essential oils, the environmental conditions in which it is found. Finally note that certain extraction procedures can alter the composition of the essential oil with respect to the vegetable of origin.

For all these reasons, essential oils are beginning to be sold including the chemotype on the label, since each chemotype will have different biological activities.

I am sorry to inform you of this issue but without knowing the chemical composition of the oils used, adequate discussions and conclusions cannot be made, despite the fact that the other studies carried out in this work are complete and interesting.

Author Response

The abstract should have the following order: background-introduction, objectives, methodology, results, discussion and conclusions. In the current form it is difficult to understand.

The abstract was reviewed and rewritten.

The introduction lacks information regarding the chemical composition and biological activity of the essential oils studied.

This information was added on "Method" section.

The method of obtaining the oils is described in lines 286 - 288.

The chemical compositions of the oils, determined by gas chromatography, are shown in table 5.

The methodology has a serious error that is difficult to solve. The essential oils used are not chemically characterized. This phase is essential for the study of biological activities. Plant essential oils depend on many factors:
In the first place would be the botanical origin, since each species has a chemical composition and even within the same species we can find several chemical races (chemotypes). The quantitative and qualitative characteristics of a species vary according to the phase of the vegetative cycle in which it is found. It also influences the qualities of essential oils, the environmental conditions in which it is found. Finally note that certain extraction procedures can alter the composition of the essential oil with respect to the vegetable of origin.

For all these reasons, essential oils are beginning to be sold including the chemotype on the label, since each chemotype will have different biological activities.

I am sorry to inform you of this issue but without knowing the chemical composition of the oils used, adequate discussions and conclusions cannot be made, despite the fact that the other studies carried out in this work are complete and interesting.

The chemical characterization of the essential oils was described on "Method" section (table 5 – Lines 286 - 285)

Round 2

Reviewer 1 Report

Dear Author

This manuscript is well revised. Great. 

Author Response

We would like to thank you for the previous recommendations for the manuscript enrichment. The final result truly pleased us. 

Reviewer 2 Report

The authors fixed main flaws, as suggested.

Still, the authors should address the following issues:

Please make appropriate correction to Table 5 related to the species included in the study (Calendula officinalis, Eucalyptus citriodora, Ricinus communis, Melaleuca alternifolia, Pinus strobus). B. virgilioides and C. longa were not investigated in the current study.

Please correct Table 5 heading – essential oil are mixtures of volatile components; from Ricinus communis seeds there was isolated fatty oil (as depicted in the table).

Please insert in the Discussion section correlations between the main constituents of the essential oils/fatty oils and their antimicrobial efficacy.

Author Response

Please make appropriate correction to Table 5 related to the species included in the study (Calendula officinalis, Eucalyptus citriodora, Ricinus communis, Melaleuca alternifolia, Pinus strobus). B. virgilioides and C. longa was not investigated in the current study.

Thank you for pointing this out. We have, accordingly, revised the table and removed the B. virgilioides and C. longa oils.

Please correct Table 5 heading – essential oil are mixtures of volatile components; from Ricinus communis seeds there was isolated fatty oil (as depicted in the table).

You raised an important point here. We have indicated that the oil samples were composed of essential oils and fatty acids. Table 5 heading as well as the manuscript text were properly corrected.

Please insert in the Discussion section correlations between the main constituents of the essential oils/fatty oils and their antimicrobial efficacy.

As suggested, we have added at discussion correlations between constituents of the oils and their antimicrobial action (lines 233 - 260).

Reviewer 3 Report

I am pleased to note that the authors have taken into account all the comments I made on the manuscript. These changes have made the work notably increase its quality.
Among the improvements made, the information on the chemical composition of the essential oils studied stands out. This information offered by the manufacturer is essential, as I already mentioned in my first review.
I think that even this information could be used to further enrich the discussion, by studying the biological activities of the major components of essential oils in a bibliographic way. But still the discussion is acceptable for Antibiotics.

Author Response

I am pleased to note that the authors have taken into account all the comments I made on the manuscript. These changes have made the work notably increase its quality.
Among the improvements made, the information on the chemical composition of the essential oils studied stands out. This information offered by the manufacturer is essential, as I already mentioned in my first review.
I think that even this information could be used to further enrich the discussion, by studying the biological activities of the major components of essential oils in a bibliographic way. But still the discussion is acceptable for Antibiotics.

We have accordingly added at discussion section correlations between constituents of the oils and their antimicrobial action (lines 233 - 260).